# Characteristics, clinical outcomes, and mortality of older adults living with HIV receiving antiretroviral treatment in the sub-urban and rural areas of northern Thailand

Linda Aurpibul[1]☯, Patumrat Sripan[1]☯, Wason Paklak[1]‡, Arunrat Tangmunkongvorakul[1]‡, Amaraporn Rerkasem[1]‡, Kittipan Rerkasem[1,2]‡, Kriengkrai Srithanaviboonchai[1,2]*

1 Research Institute for Health Sciences, Chiang Mai University, Chiang Mai, Thailand, 2 Faculty of Medicine, Chiang Mai University, Chiang Mai, Thailand

☯ These authors contributed equally to this work.
‡ WP, AT, AR and KR also contributed equally to this work.
* kriengkrai.s@cmu.ac.th

## Abstract

Since the introduction of antiretroviral treatment (ART), people living with HIV worldwide live into older age. This observational study described the characteristics, clinical outcomes, and mortality of older adults living with HIV (OALHIV) receiving ART from the National AIDS program in northern Thailand. Participants aged ≥ 50 years were recruited from the HIV clinics in 12 community hospitals. Data were obtained from medical records and face-to-face interviews. In 2015, 362 OALHIV were enrolled; their median (interquartile range) age and ART duration were 57 years (54–61), and 8.8 years (6.4–11.2), respectively. At study entry, 174 (48.1%) had CD4 counts ≥ 500 cells/mm$^3$; 357 of 358 (99.6%) with available HIV RNA results were virologic-suppressed. At the year 5 follow-up, 39 died, 11 were transferred to other hospitals, 3 were lost to follow-up, and 40 did not contribute data for this analysis, but remained in care. Among the 269 who appeared, 149 (55%) had CD4 counts ≥ 500 cells/mm$^3$, and 227/229 tested (99%) were virologic-suppressed. The probability of 5-year overall survival was 89.2% (95% confidence interval, CI 85.4–92.1%). A significantly low 5-year overall survival (66%) was observed in OALHIV with CD4 counts < 200 cells/mm$^3$ at study entry. The most common cause of death was organ failure in 11 (28%), followed by malignancies in 8 (21%), infections in 5 (13%), mental health-related conditions in 2 (5%), and unknown in 13 (33%). In OALHIV with stable HIV treatment outcomes, mortality from non-infectious causes was observed. Monitoring of organ function, cancer surveillance, and mental health screening are warranted.

## Introduction

With effective antiretroviral treatment (ART), a decrease in the mortality rate and an increasing life expectancy of people living with HIV (PLHIV) were observed [1, 2]. In Thailand, the

**Data Availability Statement:** All relevant data are within the paper and its Supporting information files.

**Funding:** KS: Chiang Mai University The funder play no role in the study design, data collection and analysis, decision to publish, or preparation of the manuscript.

**Competing interests:** The authors have declared that no competing interests exist.

universal coverage National AIDS Program provides HIV care and ART free of charge since 2007 [3]. The thresholds set for starting treatment have been updated since then, from CD4 < 200 cells/mm$^3$ during 2008–2010, CD4 < 350 cells/mm$^3$ from 2011–2013, and any CD4 level since 2014 [4]. The practice following this guideline led to an overall increase in the number of PLHIV accessing ART; in 2019, an estimated 470,000 PLHIV in Thailand were receiving treatment [5]. Like other parts of the world, there is a demographic shift in HIV care as PLHIV could live longer.

Referring to the US National Institute of Health's Office of AIDS Research, individuals aged 50 years or older are defined as older adults living with HIV (OALHIV) [6]. OALHIV were more likely to be diagnosed late due to low perception of HIV-acquisition risk, limited knowledge about the disease [7], misunderstandings about HIV/AIDS, as well as HIV-related stigmatization, and discrimination [8]. The cohort data from Public Health England mentioned late diagnosis as a strong predictor of death [1]. In addition, late ART initiation after immunosuppression is associated with a slow and low rate of CD4 recovery [9, 10].

The South African study documented a higher mortality rate in OALHIV when compared to PLHIV at a younger age while receiving ART [11]. The Ugandan study documented higher mortality of OALHIV than the age-matched population without HIV [12]. Comorbidities including cardiovascular disease, heart/vascular, and malignancy were causes of death that were strongly associated with age in the ART collaboration cohort [13]. Immune dysfunction and chronic inflammation were associated with an increased risk of malignancies [14], while the use of certain ART regimens was associated with cardiovascular risk [15]. In the US veterans' cohort, older veterans with HIV had higher odds of multimorbidity than those without HIV; while renal vascular and pulmonary diseases were associated with more advance HIV infection [16].

As a result of the age-related physiologic change, we hypothesize that the treatment outcome in Thai OALHIV might be worse than PLHIV at a younger age, and the mortality was supposed to be higher. Currently, data on OALHIV in Asia is scarce. This study aimed to describe the characteristics, clinical outcomes, and mortality of OALHIV in the rural and suburban areas in northern Thailand. The study results would enable us to gain insights into the long-term impact of the national HIV program, and guide developing of surveillance program targeted OALHIV.

## Materials and methods

### Study population

The prospective observational cohort study was started in August 2015. The study participants were OALHIV who attended HIV clinics for antiretroviral treatment at 12 community hospitals in Chiang Mai, Thailand (of the 24 community hospitals in the province). They were hospitals with high number of PLHIV receiving care and treatment ranging from 300 to 1128 at the time prior to the baseline survey when the study was started. The 30 oldest PLHIV among those aged at least 50 years old at each selected hospital were approached to create a cohort that represented aging PLHIV. For each hospital, the oldest patients were approached first. The potential participants were informed and invited to join the study during routine clinical visits on a first come first serve basis. The inclusion criteria were 1) having a diagnosis of HIV infection, 2) aged ≥ 50 years, and 3) receiving antiretroviral treatment at study enrollment. After enrollment, all participants remained followed in HIV clinics for regular care under the Thai national AIDS program, in which CD4 counts and HIV RNA testing were performed annually. ART and HIV-related care were supported by the National Health Security Office. All government hospitals follow the Thailand National Guidelines on HIV/AIDS Treatment

and Prevention [17]. We continued recruiting from this cohort for multiple follow-up (FU) activities at 1-yr FU (2016) and 5-yr FU (2020). The main cohort aims to follow the quality of life and health outcomes of OALHIV receiving the standard of HIV care in Thailand [18]. This is a substudy focusing on the treatment outcomes.

## Data collection and tools

Baseline demographic data were obtained by face-to-face interviewing with trained study staff upon their enrollment to the main cohort between August and September 2015. Variables included age, sex, years of formal education, employment status, family status, monthly income, history of smoking, and alcohol use were collected. Body weight and height were measured, and the body mass index (BMI) was calculated. The clinical information from medical records was retrieved by HIV clinic staff at each hospital including duration of treatment, ART regimen, comorbidities (hypertension, diabetes, dyslipidemia, and chronic renal failure), CD4 count, and HIV RNA level within 12 months prior to the study entry. The data were deidentified prior to review by investigators. During 1-yr FU (2016) and 5-yr FU (2020), more questionnaires and activities were added to the main study. We reconsented study participants using an updated version of the study protocol approved by the Ethics Committee. Using the study case record form, CD4 count, HIV RNA level, and comorbidities were extracted from electronic medical records by HIV clinic staff. For participants who did not show up at the clinic, their last known vital or follow-up status was recorded as dead, or alive but lost-to-follow-up, or alive with transferred out to other hospitals.

For those who died in the hospitals, the cause of death and/or final diagnosis before death were reviewed from medical records by HIV clinic staff. For participants who died outside the healthcare facilities, medical record was also reviewed for the most recent diagnosis which might be relevant to their mortality. In addition, their next-to-kin relatives were contacted by HIV clinic staff who were familiar with their families. They were asked for coming to clinic for informed consent for an interview. The verbal autopsy questionnaire adapted from the WHO2016 verbal autopsy instrument to be used as an interview guide [19]. Data from the last clinic visit, hospital records, and laboratory results were reviewed to gather as much information as possible to identify the cause of death for each participant. Data from all sources were triangulated. The immediate, contributing, and underlying causes of death were discussed. The primary causes of death were finally categorized into 4 groups: 1) Organ failure: renal, liver, heart, and respiratory failure, 2) Malignancies, 3) Infections, and 4) Mental health-related conditions, by two investigators who are medical doctors (KS and LA). In cases with insufficient data, the primary cause of death was marked as unknown.

## Ethics statement

The study was approved by the institutional review board at Research Institute for Health Sciences, Chiang Mai University (Certificate approval numbers 39/2015 and 35/2020). Written informed consent was obtained from each participant before enrollment.

## Statistical analysis

Baseline demographic characteristics of study participants were reported as medians and interquartile ranges (IQRs) for continuous variables, or numbers and percentages for categorical variables. The comparison was made between those with CD4 counts < 200, 200–499, and $\geq$ 500 cells/mm$^3$ using Kruskal-Wallis test for continuous variables and Fisher's exact test for categorical variables.

CD4 counts at 5-yr FU were categorized to 3 groups: <200, 200–499 and ≥500 cells/ mm³ which represented those with severely, mildly immunosuppressed, and without immunosuppression, respectively. The proportions of male, lifetime smoking, and alcohol use in the past year in group of CD4 were compared using Fisher's exact test. The Kruskal–Wallis Test was used to compare median age, years of formal education, duration after HIV diagnosis, duration on ART, and CD4 counts at study enrollment between the group of CD4 at the 5-yr FU

The survival rates were estimated using the Kaplan-Meier method. Overall 5-year survival was defined as the study entry to the date of death from any causes. The probabilities of 5-year overall survival were calculated for each CD4 group and compared using Log rank test. Factors associated with the probability of 5-year overall survival were determined and reported as hazard ratio (HR). CD4 counts at study entry, sex, age 5-year interval, number of comorbidities, lifetime smoking, body mass index, and living status (living alone without other family members) were included. Factors with significantly associated indicated by $p < 0.05$ or with clinically relevant were included in the adjusted model. The adjusted hazard ratio (aHR) was reported. The statistical significance was defined at $p < 0.05$, and all P- values reported in this article are two-sided values, determined using Stata version 14 (StataCorp LP, College Station, TX, USA).

## Results

### Cohort description

In 2015, a total of 364 OALHIV were approached and screened; all agreed to join, but two were excluded as they have not yet initiated on ART. Three hundred and sixty-two OALHIV were enrolled into the cohort; 155 (43%) were male. Their median age was 57 years (interquartile range, IQR 54–61). There were 262 (72.4%) who had primary school education, while 55 (15.2%) attended secondary school or higher, 45 (12.4%) have not attended school. At the time of study enrollment, 175 (48.3%) were unemployed or have retired, while 74 (20.4%) remained laborers/ employed for wages, 48 (13.3%) worked in agricultural, 12 (3.3%) in government sectors, and 53 (14.6%) had independent jobs. A hundred and thirty-four (37%) reported smoking in their lifetime, and 68 (18.8%) used alcohol in the past year. Seventy-nine (21.8%), 243 (67.1%), and 40 (11.1%) had BMI in low, normal, and overweight range, respectively.

Their median duration on ART was 8.8 years (IQR 6.4–11.2). The most common ART regimens used were non-nucleoside reverse transcriptase inhibitor (NNRTI)-based in 336 (92.8%), others 26 (7.2%) were on the protease-inhibitor (PI)-based regimens. At this study enrollment, 163 (45.0%), and 174 (48.1%) had CD4 counts 200–499, and > 500 cells/mm³, respectively; 357 of 358 (99.6%) with available HIV RNA results had virologic suppression within 12 months prior to the study enrollment. Comparison of characteristics between the group with CD4 counts < 200, 200–499, and ≥ 500 cells/mm³ are shown in Table 1.

In 5-yr FU (2020), there were 269 OALHIV who showed up; 39 died, 11 were transferred to other hospitals, 38 remained in care but were unavailable to come within the visit timeframe, 2 were bedridden (one post-operative and another post-accident), and 3 were lost-to-follow-up. (Fig 1).

### Characteristics of study participants in 5-yr FU

All OALHIV who attended the clinics during the recruitment period for 5-yr FU provided consent to participate. Of the 269 participants, 110 (41%) were male. Their current median age was 61 years (IQR 59–65). There were 149 (55%), 108 (40%), and 12 (5%) with CD4 counts ≥ 500, 200–499, and < 200 cells/mm³, respectively. The group with CD4 counts < 200 cells/mm³ were at older age (median 64 years), 10 out of 12 (83%) were male, 70% were

**Table 1. Demographic characteristics of older adults living with HIV stratified by CD4 counts at study enrollment in 2015.**

| Characteristics | | CD4 counts | | | |
|---|---|---|---|---|---|
| | Total | <200 cells/mm$^3$ | 200–499 cells/mm$^3$ | ≥500 cells/mm$^3$ | p-value[a] |
| Number of participants (%) | 362 | 25 (6.9) | 163 (45.0) | 174 (48.1) | |
| Male sex | 155 (42.8) | 18 (72.0) | 85 (52.2) | 52 (29.9) | <0.001 |
| Current age, years | 57 (54–61) | 60 (56–64) | 58 (54–62) | 56 (53–60) | 0.007 |
| Education Level | | | | | 0.049 |
| Never attended school | 45 (12.4) | 3 (12.0) | 25 (15.3) | 17 (9.8) | |
| Primary school | 262 (72.4) | 22 (88.0) | 109 (66.9) | 131 (75.3) | |
| Secondary school or higher | 55 (15.2) | 0 (0) | 29 (17.8) | 26 (14.9) | |
| Employment status | | | | | 0.502 |
| Agricultural | 48 (13.3) | 4 (16.0) | 25 (15.3) | 19 (10.9) | |
| Laborer/ employed for wages | 74 (20.4) | 6 (24.0) | 34 (20.9) | 34 (19.5) | |
| Unemployed/ retired | 175 (48.3) | 9 (36.0) | 79 (48.5) | 87 (50.0) | |
| Independent job/business | 53 (14.6) | 5 (20.0) | 18 (11.0) | 30 (17.2) | |
| Government sectors/ state enterprise | 12 (3.3) | 1 (4.0) | 7 (4.3) | 4 (2.3) | |
| Monthly income, Thai baht[b] | 6,000 | 6500 | 6167 | 6,000 | 0.690[c] |
| | (3,100–12,500) | (2,000–15,000) | (3333–15,000) | (3,200–12,000) | |
| Lifetime smoking | 134 (37.0) | 14 (56.0) | 71 (43.6) | 49 (28.2) | 0.002 |
| Alcohol intake in the past year | 68 (18.8) | 3 (12.0) | 35 (21.5) | 30 (17.2) | |
| Body mass index, kg/m$^2$ | | | | | 0.235 |
| < 18.5 (low) | 79 (21.8) | 9 (36.0) | 39 (23.9) | 31 (17.8) | |
| 18.5–24.9 (normal) | 243 (67.1) | 15 (60.0) | 105 (64.4) | 123 (70.7) | |
| ≥ 25 (overweight) | 40 (11.1) | 1 (4.0) | 19 (11.7) | 20 (11.5) | |
| Living alone without other family members | 64 (17.7) | 9 (36.0) | 22 (13.5) | 33 (19.0) | 0.026 |
| Duration of treatment, years | 8.8 (6.4–11.2) | 5.2 (2.3–8.4) | 8.3 (5.2–10.4) | 10.0 (8.1–11.8) | <0.001[b] |
| Current ART regimens | | | | | 1.000 |
| NNRTI-based | 336 (92.8) | 24 (96.0) | 151 (92.6) | 161 (92.5) | |
| PI-based | 26 (7.2) | 1 (4.0) | 12 (7.4) | 13 (7.5) | |
| Presence of comorbidities | 242 (66.8) | 19 (76.0) | 109 (66.8) | 114 (65.5) | 0.598 |
| Undetectable HIV RNA level at study enrollment[d] | 357/358 (99.7) | 24/24 (100) | 159/160 (99.4) | 174/174 (100) | 0.514 |

Data in number (%), or median (interquartile range, IQR)

kg kilograms; m meters; NNRTI non-nucleoside reverse transcriptase inhibitors; PI protease-inhibitors; ART antiretroviral treatment

Comorbidities included hypertension, dyslipidemia, diabetes, and chronic kidney disease

[a] p-value from fisher's exact test

[b] The exchange rate in September 2020, 31.586 Thai baht was equal to 1 USD

[c] p-value from Kruskal-Wallis test

[d] Four participants had no HIV RNA testing results

lifetime smokers, and had low CD4 counts (median 222 cells/mm$^3$) at the study entry. Comparison of their characteristics are shown in Table 2.

## CD4 and HIV RNA trends

The CD4 distributions of OALHIV who attended the 5-yr FU (n = 269) from the study enrollment (2015) to the 5-yr FU (2020) stratified by sex was depicted in Fig 2. During the 5-yr FU, the median CD4 counts was 484 cells/ mm$^3$ (IQR 339–634); with slightly more than half (55%) of participants had CD4 cell count ≥ 500 cells/mm$^3$. Female had a significantly higher median

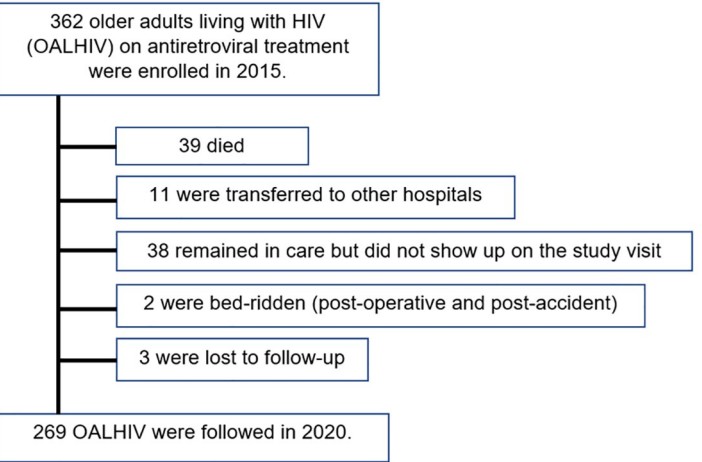

**Fig 1. Flow chart of study participants' status.**

CD4 counts when compared to male OALHIV both study entry (556 vs. 438 cells/ mm$^3$, respectively; p < 0.001) and at 5-yr FU (539 vs. 416 cells/ mm$^3$, respectively; p < 0.001). The HIV RNA level of OALHIV was illustrated in Fig 3. There was no significant difference in the percentage of virologic suppression between sexes at the study entry and during the entire follow-up period. The virologic suppression rates ranged from 93.0 to 97.8% in females and 94.2% to 100% in males.

## Overall survival and mortality

In the 5-year follow-up time after study entry, a total 39 participants died. The probability of 5-year overall survival was 89.2% (95% confidence interval, CI 85.4–92.1%). The Kaplan-

**Table 2. CD4 distribution and associated factors in older adults living with HIV at 5-year follow-up in 2020 (n = 269).**

| Characteristics | CD4 counts (cells/mm$^3$) | | | p-value[a] |
|---|---|---|---|---|
| | <200 | 200–499 | ≥500 | |
| | n = 12 (5%) | n = 108 (40%) | n = 149 (55%) | |
| Age, years | 64 (62–68) | 62 (59–66) | 61 (59–65) | 0.128 |
| Male sex | 10 (83%) | 61 (56%) | 39 (26%) | <0.001 |
| Monthly income, Thai baht | 7,000 | 3500 | 5,000 | 0.129[b] |
| | (3,000–13,500) | (2,000–7,000) | (2,500–10,000) | |
| Formal education, years | 4 (4–4) | 4 (4–4) | 4 (4–4) | 0.727 |
| Lifetime Smoking | 7 (70%) | 57 (53%) | 46 (34%) | <0.001 |
| Alcohol intake in the past years | 5 (42%) | 27 (25%) | 27 (18%) | 0.099 |
| Duration of antiretroviral treatment, years | 9.05 (6.15–12.95) | 9.80 (8.35–12.55) | 10.8 (8.6–13.9) | 0.116[b] |
| Duration after HIV diagnosis, years | 27.7 | 26.6 | 25.7 | 0.254[b] |
| | (24.1–28.6) | (24.4–29.5) | (22.8–28.5) | |
| CD4 counts at study entry, cells/mm$^3$ | 222 (183–318) | 424 (306–494) | 628 (496–816) | <0.001[b] |
| Undetectable HIV RNA level at study entry | 12 (100%) | 101(98%) | 110 (98%) | 1.000 |

Data in number (%), or median (interquartile range, IQR)

[a] p-value from fisher's exact test

[b] p-value from Kruskal-Wallis test

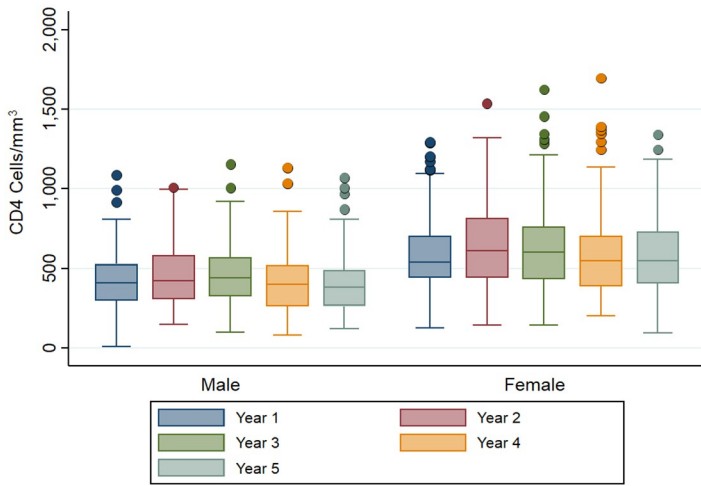

**Fig 2. CD4 distributions of older adults living with HIV at enrollment (2015) and yearly up to 5-yr FU (2020) stratified by sex.**

Meier survival estimate of all participants is shown in Fig 4. The Log-rank test revealed that OALHIV with CD4 counts $< 200$ cells/mm³ at study entry had a significant lower probability of 5-year overall survival (66.0%) when compared to other two groups with CD4 counts 200–499 cells/mm³ (89.4%), and $\geq 500$ cells/mm³ (92.5%); $p < 0.001$ (Table 3). The probabilities of 5-year overall survival in each CD4 group are shown in Fig 5.

Of those who died, 17/39 (43.5%) were female. Their median age at death was 62 years (IQR 58–67); 14 (36%), 19 (49%), and 6 (15%) were $< 60$, 60–69, and 70–79 years at the time of death, respectively. Only one participant who died had a virologic failure and septicemia was the cause of death, while other 38 OALHIV were virologic-suppressed at the most recent

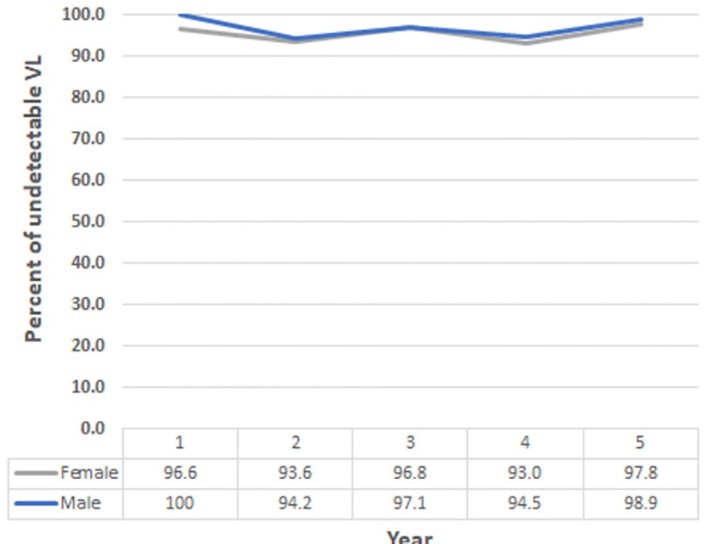

**Fig 3. Percentage of older adults living with HIV with undetectable HIV RNA level at enrollment (2015) and yearly up to 5-yr FU (2020) stratified by sex.**

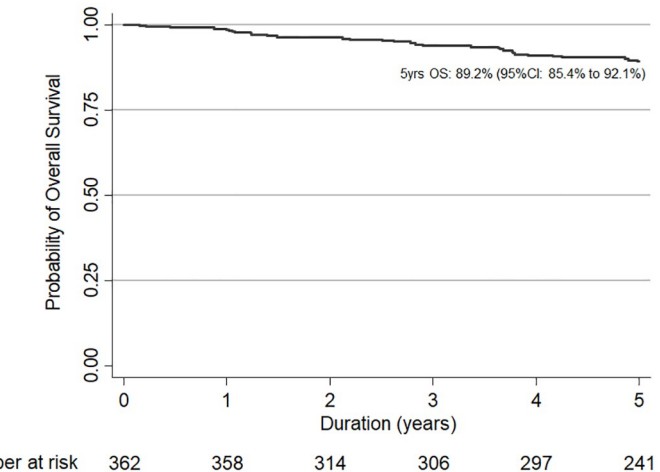

**Fig 4. The Kaplan-Meier estimates the probabilities of 5-year overall survival of all older adults living with HIV on antiretroviral treatment.**

measure prior to death. Of these 7 (18%) with virologic suppression but CD4 counts < 200 cells/mm$^3$; their causes of death were organ failure (n = 3), malignancy (n = 2), and unknown (n = 2). The causes of death were obtained from medical records in 11 (28%), from incomplete medical records and interview with relatives in 10 (26%), from interview with relatives only in 6 (15%), and not available in 12 (31%). The most common primary cause of death was organ failure in 11 (28%), followed by malignancies in 8 (21%), infections (septicemia and tuberculous peritonitis) in 5 (13%), mental health related conditions in 2 (5%), and unknown in 13 (33%). The causes of death in each age group are shown in Table 4.

**Table 3. Predictors of mortality in 5-year follow-up time among older adults living with HIV.**

| Characteristics | HR | 95% CI | p-value | aHR | 95% CI | p-value |
|---|---|---|---|---|---|---|
| CD4 counts at study entry | | | | | | |
| <200 cells/mm$^3$ | 5.45 | 2.23–13.35 | <0.001 | 4.78 | 1.73–13.17 | 0.002 |
| 200–499 cells/mm$^3$ | 1.45 | 0.69–3.07 | 0.328 | 1.49 | 0.68–3.23 | 0.316 |
| >500 cells/mm$^3$ (ref) | 1 | | | 1 | | |
| Male | 1.08 | 0.56–2.09 | 0.808 | 0.72 | 0.33–1.59 | 0.415 |
| Age 5-year interval | 1.37 | 1.08–1.74 | 0.009 | 1.23 | 0.96–1.58 | 0.097 |
| Comorbidities | | | | | | |
| 0 (ref) | 1 | | | 1 | | |
| 1 | 0.80 | 0.36–1.82 | 0.603 | 0.61 | 0.27–1.42 | 0.770 |
| ≥2 | 1.37 | 0.62–3.01 | 0.430 | 1.13 | 0.50–2.55 | 0.255 |
| Lifetime Smoking | 1.14 | 0.58–2.22 | 0.707 | 1.00 | 0.47–2.16 | 0.980 |
| Body mass index | | | | | | |
| 18.5–24.9 kg/m$^2$ (ref) | 1 | | | | | |
| <18.5 kg/m$^2$ | 1.82 | 0.89–3.69 | 0.099 | 1.37 | 0.65–2.89 | 0.410 |
| ≥ 25 kg/m$^2$ | 0.89 | 0.27–2.97 | 0.852 | 0.93 | 0.28–3.15 | 0.913 |
| Living alone without other family members | 2.01 | 0.99–4.09 | 0.053 | 1.56 | 0.73–3.33 | 0.250 |

HR hazard ratio; aHR adjusted hazard ratio; CI confidence interval

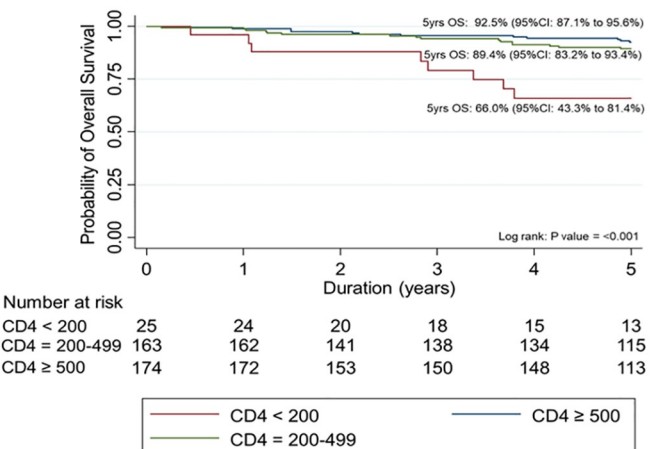

**Fig 5. The Kaplan-Meier estimates the probabilities of 5-year overall survival for older adults living with HIV on antiretroviral treatment by CD4 counts at study entry.**

## Discussion

Over the 5-year follow-up period, we found that 39 out of 362 OALHIV died, with virologic suppression in almost all but one. we documented a sustainable normalized CD4 count and virologic suppression in 95% and 99% of participants, respectively. It is worth to note that a vast majority of OALHIV in this study were initiated on ART when they were severely immunosuppressed. During the long period of living with HIV, ongoing viremia and associated inflammatory process might explain the high mortality rate [20]. According to a meta-analysis, the 10-year survival probability of progression from AID onset to AIDS-related death in PLHIV who received ART was 61% [21]. While many PLHIV might die during the first few

**Table 4. Causes of death in older adults living with HIV stratified by age at death.**

| Causes of death | 50–59 years | 60–69 years | 70–79 years | Total |
|---|---|---|---|---|
| | (n = 14) | (n = 19) | (n = 6) | (n = 39) |
| Organ failures | | | | 11 |
| Renal | 3 | 2 | | 5 |
| Liver | | 2 | | 2 |
| Heart | 1 | 1 | | 2 |
| Respiratory | | 2 | | 2 |
| Malignancies | | | | 8 |
| Lung | 1 | 1 | | 2 |
| Liver | | 1 | 1 | 2 |
| Kidney | | | 1 | 1 |
| Colon | | 1 | | 1 |
| Ovary | | 1 | | 1 |
| Unknown primary | | 1 | | 1 |
| Infections | | | | 5 |
| Sepsis | 2 | 2 | | 4 |
| TB peritonitis | | 1 | | 1 |
| Mental health-related conditions | 2 | | | 2 |
| Unknown | 5 | 4 | 4 | 13 |

years after treatment initiation, our cohort started at the median 8.8 years after ART initiation, and we documented that the probability of 5-year survival from this point was 89.2%.

We found that around half of participants had CD4 counts ≥ 500 cells/mm$^3$ at study entry and sustained up to 5-yr FU. The previous Thai study among participants with severe immunocompromised at the median age of 34.2 years reported continuously slower rate of CD4 increment after the first year on ART [9]. Principally, rapid CD4 count increase immediately after ART initiation is from cell redistribution from lymphoid tissue and peripheral memory T-cell proliferation while thymus attributed to CD4 restoration thereafter. The South African study which reported a less robust immune response in PLHIV >50 years with immunosuppression when compared to younger age groups; the mean CD4 ≥ 500 cells/mm$^3$ could be achieved at around 48 months of ART [11]. Decreased thymic function with age and low thymus reserve in older adults might cause inferior long-term CD4 restoration compared to younger age group [22]. Female OALHIV in our study had significantly higher CD4 counts than males. This was in contrast with an Ethiopian study that reported being female as a factor associated with immunologic failure in PLHIV at a younger age (80% of their patients were below 50 years) [23]. A study among PLHIV in the US and the Netherlands documented consistently lower HIV-RNA in women than men when CD4 counts were > 350 cells/mm3 [24]. We found no sex difference in HIV RNA of OALHIV in this study with high median CD4 counts.

The sustained virologic suppression observed in our study was in line with other Thai studies. The OALHIV cohort in southern Thailand reported 97.9% virologic suppression in 307 patients at 7.27 years follow-up after ART initiation [25]. Another study in Bangkok, Thailand demonstrated 97.4% virologic suppression in 340 OALHIV at 18.4 years after ART initiation [26]. Similarly favorable virologic outcome were reported from other regions. In the large French cohort, 16,436 OALHIV aged 50–74 years had a virologic suppression rate of 90.6% [27]. Another was the Cameroon study in 112 OALHIV at the average age of 57.3 years, the virologic suppression rate was 92.6% [28].

Among OALHIV in our cohort, the probability of 5-year overall survival was 89.2%. We started following OALHIV when most of them were considered as stable on treatment. Thus, it was not unexpected to see a higher probability of survival. CD4 < 200 cells/mm3 was the only associated factor to mortality. This was in line with the Ugandan study reported that CD4 counts < 100 cells/mm$^3$ was associated with high rate of death in OALHIV [29]. While only 5% of OALHIV in our study had CD4 < 200 cells/mm$^3$, 7 out of 38 (18%) OALHIV who died were virologic suppressed with CD4 counts < 200 cells/mm$^3$ at the most recent measure prior to death. Screening for organ functions, other infections, and sign/symptoms of malignancies in OALHIV with low CD4 count might be warranted.

While getting older, OALHIV could have other health challenges included non-AIDS illnesses, and age-related conditions like others without HIV at the same age range. At present, HIV RNA level is the only conventional biomarker in HIV treatment monitoring. A favorable HIV RNA only indicate low risk of HIV treatment failure but might not be sufficient, as it would not a predictive marker for overall health in ageing population. During long-term follow-up in the HIV clinic, regular screening for non-communicable diseases and malignancies remains essential in OALHIV with increasing age.

As our population were older adults with much longer duration on ART, their median age at death was 62 years. According to the World Health organization data published in 2018, the life expectancy in Thailand was 75.5 years, and HIV/AIDS as the 11[th] causes of death [30]. This was a much shorter life expectancy than general population in Thailand. The most identifiable causes of death our cohort were not AIDS, but organ failures and malignancies. The finding went with the evidence from the United Kingdom national observational cohort that

increasing proportion of death due to non-AIDS disorders such as liver disease, cardiovascular diseases, and malignancies were seen after years on treatment [1]. Data from the antiretroviral therapy cohort collaboration (ART-CC) in Europe and North America similarly reported a decrease rate of AIDS death with time since ART initiation, while the mortality from non-AIDS malignancy increased [13].

The strength of this study is that we described the treatment outcomes of OALHIV who received regular HIV care in community hospitals, while most other studies conducted in research settings or tertiary care facilities. The results might be applicable in the context of patients receiving NNRTI- or PI-based ART regimens, although not in those on INI-based or other novel regimens. There were several limitations that merit being addressed. First, we started the cohort many years after the ART initiation. Therefore, there was a possibility of survivorship bias as those who survived and remained followed were supposed to have a better outcome. Second, we introduced the study and enrolled OALHIV at each site on a first come first serve basis. It was nonrandom and did not comprehensive of the overall clinic population. Third, we did not include comparison group which could be either younger PLHIV in the same setting for the treatment outcomes, or age-matched population without HIV in the same socioeconomic environment for comorbidities. Fourth, the cause of death was unidentifiable in many cases after reviewing all available information.

In summary, OALHIV who received HIV care and ART provided by the Thai national AIDS program could survive for nearly a decade and had a stable HIV treatment outcome. Most causes of death in this population were not AIDS-related. The finding supports the need for detection of age-related conditions which could occur earlier than in general population. In those with sustain virologic suppression, monitoring of organ functions, cancer surveillance, and mental health screening should be incorporated into HIV programs for OALHIV.

## Supporting information

**S1 Dataset.**
(SAV)

## Acknowledgments

We would like to thank all older adults and their relatives who devoted time to participate in this study. We also would like to thank the study staff who conducted data collection, and the HIV clinic staff in community hospitals who helped in reviewing medical records and coordinating study activities while providing routine services.

## Author Contributions

**Conceptualization:** Linda Aurpibul, Patumrat Sripan, Wason Paklak, Arunrat Tangmunkongvorakul, Amaraporn Rerkasem, Kittipan Rerkasem, Kriengkrai Srithanaviboonchai.

**Data curation:** Linda Aurpibul, Amaraporn Rerkasem.

**Formal analysis:** Patumrat Sripan.

**Funding acquisition:** Kriengkrai Srithanaviboonchai.

**Investigation:** Linda Aurpibul, Wason Paklak, Arunrat Tangmunkongvorakul, Amaraporn Rerkasem, Kriengkrai Srithanaviboonchai.

**Methodology:** Linda Aurpibul, Patumrat Sripan, Wason Paklak, Arunrat Tangmunkongvorakul, Kriengkrai Srithanaviboonchai.

**Resources:** Arunrat Tangmunkongvorakul, Kittipan Rerkasem, Kriengkrai Srithanaviboonchai.

**Supervision:** Arunrat Tangmunkongvorakul, Kittipan Rerkasem, Kriengkrai Srithanaviboonchai.

**Validation:** Patumrat Sripan, Kriengkrai Srithanaviboonchai.

**Visualization:** Patumrat Sripan.

**Writing – original draft:** Linda Aurpibul, Patumrat Sripan.

**Writing – review & editing:** Wason Paklak, Arunrat Tangmunkongvorakul, Amaraporn Rerkasem, Kittipan Rerkasem, Kriengkrai Srithanaviboonchai.

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
