## [Decision Letter · Decision Letter 0]

28 Oct 2022

PONE-D-22-18071Characteristics, clinical outcomes, and mortality of older adults living with HIV receiving antiretroviral treatment in the sub-urban and rural areas of northern ThailandPLOS ONE

Dear Dr. Aurpibul,

Thank you for submitting your manuscript to PLOS ONE. After careful consideration, we feel that it has merit but does not fully meet PLOS ONE’s publication criteria as it currently stands. Therefore, we invite you to submit a revised version of the manuscript that addresses the points raised during the review process.

We look forward to receiving your revised manuscript.

Kind regards,

Vincent C Marconi

Academic Editor

PLOS ONE

Journal Requirements:

Reviewers' comments:

Reviewer's Responses to Questions

**Comments to the Author**

1. Is the manuscript technically sound, and do the data support the conclusions?

Reviewer #1: Yes

Reviewer #2: Partly

2. Has the statistical analysis been performed appropriately and rigorously? 

Reviewer #1: Yes

Reviewer #2: I Don't Know

3. Have the authors made all data underlying the findings in their manuscript fully available?

Reviewer #1: Yes

Reviewer #2: Yes

4. Is the manuscript presented in an intelligible fashion and written in standard English?

Reviewer #1: No

Reviewer #2: Yes

5. Review Comments to the Author

Reviewer #1: This manuscript presents valuable information namely that older patients with HIV infection, despite a very high rate of virological suppression, have increased mortality over time if the CD4 cell count is <200. However, those with CD4 of 200-499 seem to have mortality similar to those with a CD4 >500.

It also provides important data about the successful deployment of ART in older patients in Thailand with a very high rate of virologic suppression.

Introduction

The introduction is too long

Relating to this sentence: “As a result of the physiologic change from aging, we hypothesize that the HIV treatment outcome and mortality of OALHIV might be different from PLHIV at a younger age.” Was the hypothesize that older patients were merely different? In what way? Were worse ART treatment outcomes expected or better? Was the expectation about overall survival?

Methods

What role does survivorship bias possibly play here? Please discuss and acknowledge as a limitation.

Results

Do we know what proportion of patients agreed to participate versus the number approached for entry? If there was a significant rate of refusal, what does this mean in terms of whether this cohort is representative of the larger patient population.

The authors should describe how they dealt with missing viral load tests. Were patients with missing viral load data excluded from the virologic suppression analysis or was there an assumption made about their virologic outcome (for example, assumed to be suppressed)?

How do you reconcile the high rate of virologic suppression reported with the higher rate of mortality in patients with CD4<200? Were these mainly non-AIDS related deaths?

Discussion

I think the discussion should be rewritten for clarity and it should be shorter.

I don’t think you present evidence to support this sentence: “The excess mortality and morbidity could be preventable and/or manageable if detected and intervened in a timely manner.”

Figures:

I would not use more than one decimal place as this level of precision is not useful (for example, 99.59% viral suppression).

Reviewer #2: Summary

Aurpibul et al performed a descriptive analysis and a survival analysis of persons living with HIV aged >50 years in Thailand to evaluate the clinical characteristics, health outcomes, and mortality of older individuals on antiretroviral therapy in this region. ART-treated patients from 12 community based clinics were asked to participate on a first-come, first-serve basis and were followed from 2015-2020 over which time data collection occurred on intervals. Primary findings included sustained virologic suppression in vast majority of cohort (>99%), 89.2% overall survival, and primarily non-AIDS causes of death. The authors have accessed a wide collection of data in a unique setting and the focus on older adults is timely and important. However, there are several methodologic clarifications needed as outlined below and to improve readership and understanding, the manuscript could benefit from restructuring.

Major points

• The analysis focuses on older adults living with HIV (OALHIV), which the authors define as >50 yrs of age. Is this a standardized definition and if so, suggest the authors provide a citation; if not standardized, suggest the authors provide a rationale as to why this specific threshold was chosen to define older age in the context of living with HIV.

• The primary aim is described as following “quality of life and health outcomes,” however, quality of life is not actually evaluated in this analysis (in terms of using the SF-36 or other health-related quality of life scale or associated measures such as depression or anxiety, etc). Suggest the authors remove such language.

• Suggest the authors more clearly state the period of observation as being from August 2015 through November 2020 (if that’s accurate) and the frequency in which patients engaged in the research study – did this occur every 2 or 3 months as mentioned? Or only in 2015, 2016, and 2020? It is unclear at which intervals patients were followed in clinic and/or completed study visits (did these co-occur?) from 2016 through 2020, ie the visit frequency and in what capacity? Finally, perhaps it may streamline the analysis and manuscript if only 2015 and 2020 time points are included (depending on clarification from aforementioned questions). Along those lines, instead of reporting characteristics such as CD4 count at each of the three timepoints, it may be more useful to summarize the data as change in CD4 count over time from 2015 to 2020.

• In the methods description, specifically “data collection and tools,” expansion is needed as it is not clear how exactly data were collected, including which data elements, by whom, from which sources, and at what times points- authors should clarify these points precisely. Were data elements collected from paper or electronic records? Was there a case report form used? At which frequencies were data assessed? Were any data obtained by self-report (ie, is that what the survey refers to) or gathered solely from provider documentation, clinical measurements, and laboratory values, etc? What comprised a study visit? If there was a questionnaire used for patients to self-report outcomes, consider including as supplemental material; and if self-report was involved (of diagnoses or medications, etc) this should be included as a potential limitation in the discussion as self-report can be a source of recall bias.

• Along those lines, please specify in the methods how “presence of comorbidities” (as reported in tables 1 and 3) was defined – which comorbidities were included? How were comorbidities defined? How were these data ascertained, ie, by self-report of diagnosis or medication? Clinical measurement (e.g., blood pressure), and/or laboratory value (e.g., estimated GFR to define chronic kidney disease, for example).

• Suggest consistent reporting of CD4 categories as some are listed as 200-499 vs >500 cells/mm3, whereas at other times, a breakout category of 350-499 is included. Alternatively, authors should provide rationale for the 350-499 CD4 category in the more recent time point (ie, 2020). Again, consider describing CD4 trajectories instead over the entire period (as what is currently included only ranges from 2017-2020 and it is unclear why years 2015-2016 are excluded from those data).

• In the analysis reported in table 3, please include which specific variables were adjusted for in the text of the methods, and strongly suggest that sex is included in the model and reported in the table (especially considering other figures are stratified by sex); and the authors may also consider evaluating age categorically to better describe the impact of age in 5- or 10-yr increments, for example, as well as the incremental number of comorbidities in place of “presence of comorbidities,” ie, 0, 1, ≥2 comorbidities, for example. This approach may give the data more granularity and be better suited for a clinical readership with the goal of implementing findings into pare care practice.

• The authors may consider examining the impact of cumulative HIV-1 viral exposure, such as viremia copy-years, on their ascertained clinical outcomes, especially considering that longitudinal data are available for this cohort. Given that the vast majority of participants were virologically suppressed at X, this may help tease out effects of preceding intermittent or sustained periods of non-suppression, which have been associated with mortality and non-AIDS comorbidities among persons with HIV even despite treatment with ART.

Minor points

• Are there differences in the models (and associated resources) of care delivery across the 12 community sites included? Suggest the authors clarify this in the methods and if there are differences, mention in the discussion as a potential limitation. Of note, while it is mentioned that Thai national standards are followed, this is not elaborated on so perhaps that could help clarify if a reference or summary of national standards is included in an appendix.

• The concept of frailty is mentioned in the introduction and discussion, however, is not evaluated in the paper. Consider better integrating the concept of fragility into the discussion of authors’ principle findings or perhaps remove altogether.

• How was “organ failure” defined? Also could some causes of death be overlapping, such that a patient died of renal failure but also had malignancy? Suggest the authors clarify this point on causes of death.

• Please spell out “overall survival” as “OS” is not a commonly used abbreviation.

• In lines 80-82, please clarify if comorbidity prevalence is higher among those with INCREASED HIV disease severity, or rather, if higher comorbidity prevalence INCREASES HIV disease severity.

• First noted in line 102, please change “HIV-infected patients” to person-first language such as persons (living) with HIV.

• In line 109, an inclusion criteria is listed as “receiving ART at study enrollment.” Was any preceding time period of ART receipt or virologic suppression required (ie, ≥12 months for example)? It is later mentioned that patients were excluded if there was not a viral load measurement in the past 12 months, however, this is not clearly stated in the study criteria.

• In line 121, may consider “the data were deidentified prior to review by investigators.”

• In line 123, could the authors elaborate on why reconsenting was necessary at each study visit? Was the period of follow up time not determined a priori?

• In lines 136-137, what is meant by “the verbal autopsy questionnaire was used as a guide”?

• In place of lines 188-190, suggest the authors include in the study flow diagram as this exclusion of 4 patients does likely not require text. This could also be of benefit for lines 201-206, and as previously mentioned, may consider removing the interim time point of 2016 anyway as it is unclear what value it adds (as opposed to being clearer about the study observation period from 2015-2020 overall). Also, Figure 1 is not referenced in the text of the manuscript.

• In terms of causes of death, could the authors specify whether “infections,” were AIDS-related (i.e., opportunistic) vs HBV/HCV-related vs other?

• In table 4, consider totaling each row in the total column, ie, there were 5 total deaths from renal failure.

• What is the difference in Figure 3 and Figure 4? Also given that HIV suppression is >90% in this cohort, there is not much additive value in including Figures 3 and 4.

• Why are Figures 2-4 stratified by sex? Suggest providing rationale.

• In the discussion, use of immune “recovery,” may not be appropriate given that patients had been on ART for 8 years – please clarify. Along those lines, the paper may benefit from focused attention in the first paragraph of the discussion on the primary findings of paper, ie, overall health outcomes including death as opposed to CD4 trajectories.

• Authors should include in their discussion of limitations, the bias that is introduced in approaching patients to participate- it was nonrandom and also not comprehensive of the clinic population overall.

• Suggest the authors include in their discussion a mention of potential implications of their findings in the context of patients being on NNRTI- or PI-based anchor regimens, but not including INSTI-based regimens.

• References could be more up to date (many of the publications cited <2010 have corresponding data published in the last decade)

6. PLOS authors have the option to publish the peer review history of their article (what does this mean?). If published, this will include your full peer review and any attached files.

Reviewer #1: No

Reviewer #2: No

---

## [Author Response · Author response to Decision Letter 0]

28 Nov 2022

Response to reviewers’ comments

Reviewer #1

This manuscript presents valuable information namely that older patients with HIV infection, despite a very high rate of virological suppression, have increased mortality over time if the CD4 cell count is <200. However, those with CD4 of 200-499 seem to have mortality similar to those with a CD4 >500. 

It also provides important data about the successful deployment of ART in older patients in Thailand with a very high rate of virologic suppression.

Introduction

1. The introduction is too long

Response: The introduction has been revised and shortened as recommended.

2. Relating to this sentence: “As a result of the physiologic change from aging, we hypothesize that the HIV treatment outcome and mortality of OALHIV might be different from PLHIV at a younger age.” Was the hypothesis that older patients were merely different? In what way? Were worse ART treatment outcomes expected or better? Was the expectation about overall survival?

Response: We have revised the sentence in response to the comments and questions as follows.

“We hypothesize that the treatment outcome in Thai OALHIV would be worse than PLHIV at a younger age due to age-related physiological changes and that mortality would be higher.”

Methods

3. What role does survivorship bias possibly play here? Please discuss and acknowledge as a limitation.

Response: Thank you. We mentioned this possibility in page 17, the 4th paragraph of the discussion as follows:

“We started following OALHIV when most of them were considered stable on treatment. Thus, it was not unexpected to see a higher rate of survival.”

AND we also added the study limitation as follows.

“First, we started the cohort many years after ART initiation. Therefore, there was a possibility of survivorship bias as those who survived and remained followed were likely to have better outcomes.”

Results

4. Do we know what proportion of patients agreed to participate versus the number approached for entry? If there was a significant rate of refusal, what does this mean in terms of whether this cohort is representative of the larger patient population.

Response: All potential participants who were approached agreed to join the study.

We added this information in the results, page 8, as follows.

“In 2015, a total of 364 OALHIV were approached and screened. All agreed to join, but two were excluded as they had not yet initiated ART.”

AND also in page 11,

“All OALHIV who attended the clinics during the recruitment period for FU2 provided consent to participate.”

5. The authors should describe how they dealt with missing viral load tests. Were patients with missing viral load data excluded from the virologic suppression analysis or was there an assumption made about their virologic outcome (for example, assumed to be suppressed)? 

Response: There were only 4 out of 362 (1%) patients who had no data of VL. They were excluded from the virologic suppression analysis. We have added denominators and a footnote in Table 1 to make it clear. 

6. How do you reconcile the high rate of virologic suppression reported with the higher rate of mortality in patients with CD4<200? Were these mainly non-AIDS-related deaths?

Response: We added the causes of death in this part of the results to clarify this issue as follows.

“Only one deceased participant had a virologic failure, with septicemia as the cause of death, while 38 OALHIV were virologic-suppressed at the most recent measurement prior to death. Of these 38, 7 (18%) had virologic suppression with CD4 counts < 200 cells/mm3. For these 7 the causes of death were organ failure (n=3), malignancy (n=2), and unknown (n=2).”

Discussion

7. I think the discussion should be rewritten for clarity and it should be shorter.

Response: The discussion has been revised and shortened as suggested.

8. I don’t think you present evidence to support this sentence: “The excess mortality and morbidity could be preventable and/or manageable if detected and intervened in a timely manner.”

Response: Thank you. The sentence has been removed.

Figures:

9. I would not use more than one decimal place as this level of precision is not useful (for example, 99.59% viral suppression).

Response: We have modified Figure 3 as suggested. 

Reviewer #2: Summary

Aurpibul et al performed a descriptive analysis and a survival analysis of persons living with HIV aged >50 years in Thailand to evaluate the clinical characteristics, health outcomes, and mortality of older individuals on antiretroviral therapy in this region. ART-treated patients from 12 community-based clinics were asked to participate on a first-come, first-serve basis and were followed from 2015-2020 over which time data collection occurred on intervals. Primary findings included sustained virologic suppression in vast majority of cohort (>99%), 89.2% overall survival, and primarily non-AIDS causes of death. The authors have accessed a wide collection of data in a unique setting and the focus on older adults is timely and important. However, there are several methodologic clarifications needed as outlined below and to improve readership and understanding, the manuscript could benefit from restructuring.

Major points

10. The analysis focuses on older adults living with HIV (OALHIV), which the authors define as >50 years of age. Is this a standardized definition and if so, suggest the authors provide a citation; if not standardized, suggest the authors provide a rationale as to why this specific threshold was chosen to define older age in the context of living with HIV.

Response: PLHIV who are fifty years old or older are generally considered as the OALHIV. We added a citation for the age cut-off. 

“Guidelines for the Use of Antiretroviral Agents in Adults and Adolescents with HIV: Considerations for Antiretroviral Use in Special Patient Populations 2019 [Available from: https://clinicalinfo.hiv.gov/en/guidelines/hiv-clinical-guidelines-adult-and-adolescent-arv/treatment-goals?view=full.”

11. The primary aim is described as following “quality of life and health outcomes,” however, quality of life is not actually evaluated in this analysis (in terms of using the SF-36 or other health-related quality of life scale or associated measures such as depression or anxiety, etc). Suggest the authors remove such language.

Response: We have revised the text and added a published paper from this cohort as a reference to make it clear as follows.

“The main cohort aims to follow the quality of life and health outcomes of OALHIV receiving the standard of HIV care in Thailand (18). This is a sub-study focusing on the treatment outcomes.”

12. Suggest the authors more clearly state the period of observation as being from August 2015 through November 2020 (if that’s accurate) and the frequency in which patients engaged in the research study – did this occur every 2 or 3 months as mentioned? Or only in 2015, 2016, and 2020? It is unclear at which intervals patients were followed in clinic and/or completed study visits (did these co-occur?) from 2016 through 2020, i.e. the visit frequency and in what capacity? Finally, perhaps it may streamline the analysis and manuscript if only 2015 and 2020 time points are included (depending on clarification from the aforementioned questions). Along those lines, instead of reporting characteristics such as CD4 count at each of the three time points, it may be more useful to summarize the data as a change in CD4 count over time from 2015 to 2020.

Response: We revised the text to make it clearer as follows.

“After enrollment, all participants were followed in HIV clinics for regular care under the Thai national AIDS program, in which CD4 counts and HIV RNA testing were performed annually. ART and HIV-related care were supported by the National Health Security Office. All government hospitals follow the Thailand National Guidelines on HIV/AIDS Treatment and Prevention (17). We continued recruiting from this cohort for multiple follow-up (FU) activities at FU1 (2016) and FU2 (2020).”

13. In the methods description, specifically “data collection and tools,” expansion is needed as it is not clear how exactly data were collected, including which data elements, by whom, from which sources, and at what times points- authors should clarify these points precisely. Were data elements collected from paper or electronic records? Was there a case report form used? At which frequencies were data assessed? Were any data obtained by self-report (ie, is that what the survey refers to) or gathered solely from provider documentation, clinical measurements, and laboratory values, etc? What comprised a study visit? If there was a questionnaire used for patients to self-report outcomes, consider including as supplemental material; and if self-report was involved (of diagnoses or medications, etc) this should be included as a potential limitation in the discussion as self-report can be a source of recall bias.

Response: We modified the data collecting and tools as follows. For this study, only demographic information was asked from participants. Other variables were extracted from medical records by HIV clinic staff using the case record form. 

“Baseline demographic data were obtained via face-to-face interviews with trained study staff upon the subjects’ enrollment to the main cohort between August and September 2015. Variables included age, sex, years of formal education, employment status, family status, monthly income, history of smoking, and alcohol use were collected. Body weight and height were measured, and the body mass index (BMI) was calculated. The clinical information from medical records was retrieved by HIV clinic staff at each hospital including duration of treatment, ART regimen, comorbidities (hypertension, diabetes, dyslipidemia, and chronic renal failure), CD4 count, and HIV RNA level within 12 months prior to the study entry. The data were deidentified prior to review by investigators. During FU1 (2016) and FU2 (2020), more questionnaires and activities were added to the main study. We reconsented study participants using an updated version of the study protocol approved by the Ethics Committee. Using the study case record form, CD4 count, HIV RNA level, and comorbidities were extracted from electronic medical records by HIV clinic staff. For participants who did not complete all follow-ups, their last known vital or follow-up status was recorded as dead, as alive but lost-to-follow-up, or as alive but transferred out to other hospitals.”

14. Along those lines, please specify in the methods how “presence of comorbidities” (as reported in tables 1 and 3) was defined – which comorbidities were included? How were comorbidities defined? How were these data ascertained, i.e., by self-report of diagnosis or medication? Clinical measurement (e.g., blood pressure), and/or laboratory value (e.g., estimated GFR to define chronic kidney disease, for example).

Response: The presence of comorbidities relied upon documented diagnosis in electronic medical records at each hospital. We specified in the method session as follows.

“Using the study case record form, CD4 count, HIV RNA level, and comorbidities were extracted from electronic medical records by HIV clinic staff.”

15. Suggest consistent reporting of CD4 categories as some are listed as 200-499 vs >500 cells/mm3, whereas at other times, a breakout category of 350-499 is included. Alternatively, authors should provide rationale for the 350-499 CD4 category in the more recent time point (ie, 2020). Again, consider describing CD4 trajectories instead over the entire period (as what is currently included only ranges from 2017-2020 and it is unclear why years 2015-2016 are excluded from those data).

Response: We revised the CD4 categories to <200, 200-499, and ≥500 cells/mm3 in the results text, and tables 2 & 3.

“There were 12 (5%), 108 (40%), and 149 (55%) with CD4 counts < 200, 200-499, and ≥ 500 cells/mm3, respectively.”

As CD4 and VL were performed once a year. They were tested at different months of the year for each participant. We collected data from 5 consecutive years. Figure 2 was revised to include all CD4 counts from the study entry to year 5.

16. In the analysis reported in table 3, please include which specific variables were adjusted for in the text of the methods, and strongly suggest that sex is included in the model and reported in the table (especially considering other figures are stratified by sex); and the authors may also consider evaluating age categorically to better describe the impact of age in 5- or 10-yr increments, for example, as well as the incremental number of comorbidities in place of “presence of comorbidities,” ie, 0, 1, ≥2 comorbidities, for example. This approach may give the data more granularity and be better suited for a clinical readership with the goal of implementing findings into pare care practice.

Response: We have done the analysis as suggested and revised Table 3 accordingly. 

The methods part was revised to include adjusted variables as follows.

“Factors associated with the probability of 5-year OS were determined and reported as hazard ratio (HR). CD4 counts at study entry, sex, age 5-year interval, number of comorbidities, lifetime smoking, body mass index, and living status were included.”

17. The authors may consider examining the impact of cumulative HIV-1 viral exposure, such as viremia copy-years, on their ascertained clinical outcomes, especially considering that longitudinal data are available for this cohort. Given that the vast majority of participants were virologically suppressed at X, this may help tease out the effects of preceding intermittent or sustained periods of non-suppression, which have been associated with mortality and non-AIDS comorbidities among persons with HIV even despite treatment with ART.

Response: 357 out of 358 participants with available VL results (99.7%) were virologic suppressed at the study entry. As there was only 1 participant with non-suppressed VL at baseline, few of them in subsequent years, and no participant with sustain period of non-suppression, we did not perform further analyses on its effect.

Minor points

18. Are there differences in the models (and associated resources) of care delivery across the 12 community sites included? Suggest the authors clarify this in the methods and if there are differences, mention in the discussion as a potential limitation. Of note, while it is mentioned that Thai national standards are followed, this is not elaborated on so perhaps that could help clarify if a reference or summary of national standards is included in an appendix.

Response: In general, they are not different. We added more clarification and a reference in the method part as follows.

“ART and HIV-related care were supported by the National Health Security Office. All government hospitals follow the Thailand National Guidelines on HIV/AIDS Treatment and Prevention.”

19. The concept of frailty is mentioned in the introduction and discussion, however, is not evaluated in the paper. Consider better integrating the concept of fragility into the discussion of authors’ principle findings or perhaps remove altogether.

Response: We decided to remove the sentence that mentioned frailty. 

20. How was “organ failure” defined? Also could some causes of death be overlapping, such that a patient died of renal failure but also had malignancy? Suggest the authors clarify this point on causes of death.

Response: We added the definition of organ failure and revised the methods to describe clearly how the primary causes of death were determined.

“Data from all sources were triangulated. The immediate, contributing, and underlying causes of death were discussed. The primary causes of death were finally categorized into 4 groups: 1) Organ failure: renal, liver, heart, and respiratory failure, 2) Malignancies, 3) Infections, and 4) Mental health-related conditions, by two investigators who are medical doctors (KS and LA).”

21. Please spell out “overall survival” as “OS” is not a commonly used abbreviation.

Response: We have replaced all “OS” with “overall survival” as suggested.

22. In lines 80-82, please clarify if comorbidity prevalence is higher among those with INCREASED HIV disease severity, or rather, if higher comorbidity prevalence INCREASES HIV disease severity.

Response: We revised the sentence (now lines 68-71) as follows.

“In the US veterans’ cohort, older veterans with HIV had higher odds of multimorbidity than those without HIV; while renal vascular and pulmonary diseases were associated with more advance HIV infection (16).”

23. First noted in line 102, please change “HIV-infected patients” to person-first language such as persons (living) with HIV.

Response: Thank you. We now changed it to “PLHIV”.

24. In line 109, an inclusion criterion is listed as “receiving ART at study enrollment.” Was any preceding time period of ART receipt or virologic suppression required (ie, ≥12 months for example)? It is later mentioned that patients were excluded if there was not a viral load measurement in the past 12 months, however, this is not clearly stated in the study criteria.

Response: The inclusion criteria for enrollment for the main study did not specify the duration of ART prior to enrollment. All 362 participants were included in the total analysis. The viral loads of 358 who were tested within the past 12 months were included as a baseline VL for this cohort. We made it clear as follows.

“…357 of 358 (99.6%) with available HIV RNA results within 12 months prior to the study enrollment had virologic suppression.” 

25. In line 121, may consider “the data were deidentified prior to review by investigators.”

Response: We change the sentence (line 109) as suggested.

“The data were deidentified prior to review by investigators.”

26. In line 123, could the authors elaborate on why reconsenting was necessary at each study visit? Was the period of follow-up time not determined a priori?

Response: We added more clarification to make it clear about the need for reconsent as follows. The period of follow-up time was planned but encountered a delay while waiting for additional study funding. 

“During FU1 (2016) and FU2 (2020), more questionnaires and activities were added to the main study. We reconsented study participants using an updated version of the study protocol approved by the Ethics Committee.”

27. In lines 136-137, what is meant by “the verbal autopsy questionnaire was used as a guide”?

Response: We added more clarification as follows.

“The verbal autopsy questionnaire adapted from the WHO2016 verbal autopsy instrument to be used as an interview guide (19).”

28. In place of lines 188-190, suggest the authors include in the study flow diagram as this exclusion of 4 patients does likely not require text. This could also be of benefit for lines 201-206, and as previously mentioned, may consider removing the interim time point of 2016 anyway as it is unclear what value it adds (as opposed to being clearer about the study observation period from 2015-2020 overall). Also, Figure 1 is not referenced in the text of the manuscript.

Response: The flow chart has been revised and referred to in the rewritten text as followed.

“In FU2 (2020), there were 269 OALHIV who showed up; 39 died, 11 were transferred to other hospitals, 38 remained in care but were unavailable to come within the visit timeframe, 2 were bedridden (one post-operative and another post-accident), and 3 were lost-to-follow-up. (Fig 1)”

29. In terms of causes of death, could the authors specify whether “infections,” were AIDS-related (i.e., opportunistic) vs HBV/HCV-related vs other?

Response: We specified the type of infection in Table 4 and added them in the text as follows. 

“The most common primary cause of death was organ failures in 11 (28%), followed by malignancies in 8 (21%), infections (septicemia and tuberculous peritonitis) in 5 (13%),…”

30. In table 4, consider totaling each row in the total column, ie, there were 5 total deaths from renal failure.

Response: The total number for each row was added as suggested. 

31. What is the difference between Figure 3 and Figure 4? Also given that HIV suppression is >90% in this cohort, there is not much additive value in including Figures 3 and 4.

Response: Thank you for pointing this out. We are now attaching a correct Figure 4.

32. Why are Figures 2-4 stratified by sex? Suggest providing a rationale.

Response: We stratified by sex as females had a significantly higher CD4 count at both entry and FU3. Thus, the figures showed that the immunologic outcome was better in females while the virologic outcome was not different. 

33. In the discussion, use of immune “recovery,” may not be appropriate given that patients had been on ART for 8 years – please clarify. Along those lines, the paper may benefit from focused attention in the first paragraph of the discussion on the primary findings of paper, i.e., overall health outcomes including death as opposed to CD4 trajectories.

Response: We modified the term to “sustainable normalized CD4 count”.

34. Authors should include in their discussion of limitations, the bias that is introduced in approaching patients to participate- it was nonrandom and also not comprehensive of the clinic population overall.

Response: We addressed this issue in the limitation as follows.

“Second, we introduced the study and enrolled OALHIV at each site on a first come first serve basis. It was nonrandom and did not comprehensive of the overall clinic population.”

35. Suggest the authors include in their discussion a mention of potential implications of their findings in the context of patients being on NNRTI- or PI-based anchor regimens, but not including INSTI-based regimens.

Response: We added this issue in the study limitation as follows.

“The results might be applicable only in the context of patients receiving NNRTI- or PI-based ART regimens, but not those on INI-based or other novel regimens.”

36. References could be more up-to-date (many of the publications cited <2010 have corresponding data published in the last decade)

Response: We added a few more up-to-date references as suggested.

---

## [Decision Letter · Decision Letter 1]

12 Dec 2022

PONE-D-22-18071R1Characteristics, clinical outcomes, and mortality of older adults living with HIV receiving antiretroviral treatment in sub-urban and rural areas of northern ThailandPLOS ONE

Dear Dr. Aurpibul,

Thank you for submitting your manuscript to PLOS ONE. After careful consideration, we feel that it has merit but does not fully meet PLOS ONE’s publication criteria as it currently stands. Therefore, we invite you to submit a revised version of the manuscript that addresses the points raised during the review process.

We look forward to receiving your revised manuscript.

Kind regards,

Vincent C Marconi

Academic Editor

PLOS ONE

Journal Requirements:

Reviewers' comments:

Reviewer's Responses to Questions

**Comments to the Author**

1. If the authors have adequately addressed your comments raised in a previous round of review and you feel that this manuscript is now acceptable for publication, you may indicate that here to bypass the “Comments to the Author” section, enter your conflict of interest statement in the “Confidential to Editor” section, and submit your "Accept" recommendation.

Reviewer #1: All comments have been addressed

Reviewer #2: (No Response)

2. Is the manuscript technically sound, and do the data support the conclusions?

Reviewer #1: Yes

Reviewer #2: Yes

3. Has the statistical analysis been performed appropriately and rigorously? 

Reviewer #1: I Don't Know

Reviewer #2: Yes

4. Have the authors made all data underlying the findings in their manuscript fully available?

Reviewer #1: Yes

Reviewer #2: Yes

5. Is the manuscript presented in an intelligible fashion and written in standard English?

Reviewer #1: Yes

Reviewer #2: No

6. Review Comments to the Author

Reviewer #1: The authors have fully and convincingly responded to my comments. I recommend acceptance of the revised manuscript.

Reviewer #2: Summary

Aurpibul et al performed a descriptive analysis and a survival analysis of persons living with HIV aged >50 years in Thailand to evaluate the clinical characteristics, health outcomes, and mortality of older individuals on antiretroviral therapy in this region. The authors did an excellent job revising the paper and responding to reviewer comments. A few comments below for additional clarity that would improve readability and refine scope of this manuscript prior to publication. The analysis remains an important contribution to the literature on health outcomes among older adults in a unique global setting.

Remaining points

• Intro- lines 55-56, U.S. CDC is mentioned, however, reference corresponds with HIV.gov without clear reference to CDC, suggest reconciling (cite CDC directly or reference in manuscript text as Office of AIDS research/ HIV.gov)

• FU1 and FU2 is a helpful notation of the follow-up time, however, would the authors consider instead “1-yr follow-up” and “5-yr follow-up” so that FU2 isn’t confused as two years instead of 5?

• Line 153 contains “OS” instead of overall survival, please make sure all abbreviations of OS were removed for clarity

• Line 155- what is meant by “living status”?

• Line 215- study entry CD4 is listed, is this only among the 269 with 5yr follow up however? Suggest authors clarify this point in the subsection.

• The authors report in Result text and figures significant sex differences in study entry and end of follow-up CD4 but no significant differences by sex in HIV RNA at 5yr follow up- what about study entry (since this was mentioned for CD4)? Perhaps that is the 93-100% range, however, unclear if that range is for men/women or FU1/FU2? Also, given that authors spend text and figures on sex stratification, strongly suggest they include a synopsis in the discussion about why they evaluated such findings by sex and any hypotheses on why differences were or were not observed by sex.

• Table 4- please list the number of participants as a column header for each column (instead of total in the bottom row)

• Of note, the automated abstract did not match that in the word doc and it seems that included in the word doc is updated (notable edits made and appreciated)

7. PLOS authors have the option to publish the peer review history of their article (what does this mean?). If published, this will include your full peer review and any attached files.

Reviewer #1: No

Reviewer #2: No

---

## [Author Response · Author response to Decision Letter 1]

19 Dec 2022

Reviewer #1: The authors have fully and convincingly responded to my comments. I recommend acceptance of the revised manuscript.

Response: Thank you for your review

Reviewer #2: Summary

Aurpibul et al performed a descriptive analysis and a survival analysis of persons living with HIV aged >50 years in Thailand to evaluate the clinical characteristics, health outcomes, and mortality of older individuals on antiretroviral therapy in this region. The authors did an excellent job revising the paper and responding to reviewer comments. A few comments below for additional clarity that would improve readability and refine scope of this manuscript prior to publication. The analysis remains an important contribution to the literature on health outcomes among older adults in a unique global setting.

Remaining points

• Intro- lines 55-56, U.S. CDC is mentioned, however, the reference corresponds with HIV.gov without clear reference to CDC, suggest reconciling (cite CDC directly or reference in manuscript text as Office of AIDS research/ HIV.gov)

Response: We changed the manuscript text as suggested.

“Referring to the US National Institute of Health’s Office of AIDS Research, individuals aged 50 years or older are defined as older adults living with HIV (OALHIV)”

• FU1 and FU2 is a helpful notation of the follow-up time, however, would the authors consider instead “1-yr follow-up” and “5-yr follow-up” so that FU2 isn’t confused as two years instead of 5?

Response: Thank you. We changed FU1 to 1-yr FU and FU2 to 5-yr FU as suggested.

• Line 153 contains “OS” instead of overall survival, please make sure all abbreviations of OS were removed for clarity

Response: We apologized for the missing. All “OS” in the abstract and manuscript were now changed to “overall survival”. 

• Line 155- what is meant by “living status”?

Response: We clarified the term “living status” in the bracket as follows.

‘living status (living alone without other family members)”

• Line 215- study entry CD4 is listed, is this only among the 269 with 5yr follow up however? Suggest authors clarify this point in the subsection.

Response: We clarified in the first sentence of this subsection as follows.

“The CD4 distributions of OALHIV who attended the 5-yr FU (n=269) from the study enrollment (2015) to the 5-yr FU (2020) stratified by sex was depicted in Fig 2.”

• The authors report in Result text and figures significant sex differences in study entry and end of follow-up CD4 but no significant differences by sex in HIV RNA at 5yr follow up- what about study entry (since this was mentioned for CD4)? Perhaps that is the 93-100% range, however, unclear if that range is for men/women or FU1/FU2? Also, given that authors spend text and figures on sex stratification, strongly suggest they include a synopsis in the discussion about why they evaluated such findings by sex and any hypotheses on why differences were or were not observed by sex.

Response: We have revised the text as follows.

“There was no significant difference in the percentage of virologic suppression between sexes at the study entry and during the entire follow-up period. The virologic suppression rates ranged from 93.0 to 97.8% in females and 94.2% to 100% in males.” 

For the discussion, we included the sex difference from other studies and made a comparison to our findings as follows.

“Female OALHIV in our study had significantly higher CD4 counts than males. This was in contrast with an Ethiopian study that reported being female as a factor associated with immunologic failure in PLHIV at a younger age (80% of their patients were below 50 years). A study among PLHIV in the US and the Netherlands documented consistently lower HIV-RNA in women than men when CD4 counts were > 350 cells/mm3. We found no sex difference in HIV RNA of OALHIV in this study with high median CD4 counts.”

• Table 4- please list the number of participants as a column header for each column (instead of the total in the bottom row)

Response: Thank you. We made the change as suggested. 

• Of note, the automated abstract did not match that in the word doc and it seems that included in the word doc is updated (notable edits made and appreciated)

Response: Thank you. We have included the revised abstract in this submission.

---

## [Editor Report · Decision Letter 2]

1 Mar 2023

Characteristics, clinical outcomes, and mortality of older adults living with HIV receiving antiretroviral treatment in the sub-urban and rural areas of northern Thailand

PONE-D-22-18071R2

Dear Dr. Aurpibul,

We’re pleased to inform you that your manuscript has been judged scientifically suitable for publication and will be formally accepted for publication once it meets all outstanding technical requirements.

Kind regards,

Vincent C Marconi

Academic Editor

PLOS ONE
---

## [Editor Report · Acceptance letter]

13 Mar 2023

PONE-D-22-18071R2 

Characteristics, clinical outcomes, and mortality of older adults living with HIV receiving antiretroviral treatment in the sub-urban and rural areas of northern Thailand 

Dear Dr. Aurpibul:

I'm pleased to inform you that your manuscript has been deemed suitable for publication in PLOS ONE. Congratulations! Your manuscript is now with our production department. 

Kind regards, 

on behalf of

Dr. Vincent C Marconi 

Academic Editor

PLOS ONE